# Decline in Iran's groundwater recharge

Roohollah Noori [1,2] ✉, Mohsen Maghrebi[1], Søren Jessen [3], Sayed M. Bateni[4], Essam Heggy [5,6], Saman Javadi[7], Mojtaba Noury [8], Severin Pistre [9], Soroush Abolfathi [10] & Amir AghaKouchak [11,12,13]

Groundwater recharge feeds aquifers supplying fresh-water to a population over 80 million in Iran—a global hotspot for groundwater depletion. Using an extended database comprising abstractions from over one million groundwater wells, springs, and qanats, from 2002 to 2017, here we show a significant decline of around −3.8 mm/yr in the nationwide groundwater recharge. This decline is primarily attributed to unsustainable water and environmental resources management, exacerbated by decadal changes in climatic conditions. However, it is important to note that the former's contribution outweighs the latter. Our results show the average annual amount of nationwide groundwater recharge (i.e., ~40 mm/yr) is more than the reported average annual runoff in Iran (i.e., ~32 mm/yr), suggesting the surface water is the main contributor to groundwater recharge. Such a decline in groundwater recharge could further exacerbate the already dire aquifer depletion situation in Iran, with devastating consequences for the country's natural environment and socio-economic development.

Securing groundwater is essential since it globally supplies approximately half of the irrigation water and most of the domestic water demands[1]. Given that 60% of the global population resides in regions that are expected to experience water scarcity and reduction in clean surface water availability by 2050[2], reliance on groundwater resources will inevitably further increase in the future. However, despite the importance of groundwater as one of the pillars of food and water security, extractions from aquifer systems exceed their natural recharge, leading to the rapid depletion of aquifers across the world[3–5]. Therefore, re-establishing the balance between the amount of groundwater withdrawal and recharge is essential to sustainably use groundwater resources, especially in (semi)arid regional scales such as Iran, which is known as a global hotspot for aquifers depletion[6–8].

The imbalance between groundwater withdrawal and recharge (i.e., groundwater depletion) was first reported in some aquifers in Iran in the 1970s. However, unpublished national observations revealed groundwater depletion in some plains from as early as the 1950s. This coincided with the gradual replacement of Persian qanats, which were sustainable groundwater extraction systems and UNESCO World Cultural Heritage sites[9], with (semi)deep wells. This shift was a consequence of the modernization in the agricultural sector[8]. Currently, Iran records the volume of groundwater withdrawal from over one million extraction points, including wells, springs, and qanats, through its national groundwater monitoring network that covers all 30 national hydrological basins (Fig. S1). However, obtaining ground-truth data on nationwide groundwater recharge remains challenging due to the site-specific nature of measurements and the complex interaction between various hydroclimatic and physiographic factors. Factors such as precipitation, evapotranspiration, topography, depth to the groundwater table, geology, and land use/cover (LUC) play a crucial

[1]Graduate Faculty of Environment, University of Tehran, Tehran, Iran. [2]Faculty of Governance, University of Tehran, Tehran, Iran. [3]Department of Geosciences and Natural Resource Management, University of Copenhagen, Copenhagen, Denmark. [4]Department of Civil and Environmental Engineering and Water Resources Research Center, University of Hawaii at Manoa, Honolulu, HI, USA. [5]Viterbi School of Engineering, University of Southern California, Los Angeles, CA, USA. [6]Jet Propulsion Laboratory, California Institute of Technology, Pasadena, CA, USA. [7]Department of Water Engineering, College of Abouraihan, University of Tehran, Tehran, Iran. [8]Iran Water Resources Management Company, Ministry of Energy, Tehran, Iran. [9]HydroSciences Montpellier, University of Montpellier, CNRS, IRD, Montpellier, France. [10]School of Engineering, University of Warwick, Coventry, UK. [11]Department of Civil and Environmental Engineering, University of California, Irvine, USA. [12]Department of Earth System Science, University of California, Irvine, USA. [13]Institute for Water, Environment and Health, United Nations University, Hamilton, ON, Canada. ✉e-mail: noor@ut.ac.ir

role in this process[10,11]. In fact, spatial and temporal variability of groundwater recharge, as an index of groundwater availability[12], is not accessible even locally in most parts of the world, including Iran. This scarcity of data indicates how rarely this variable is monitored. A possible way to obtain estimates of groundwater recharge over a large spatial scale is using global/regional hydrologic models. However, the recharge estimated by these models is not certain, especially in semi-arid regions such as Iran, where uncertainties in evapotranspiration and precipitation may lead to large errors in the results[13,14], hindering the detection of regional imbalances between groundwater recharge and human withdrawals[8].

Following the study conducted by Noori et al.[8] on Iran's groundwater depletion, which raised concerns about the lack of the country's groundwater recharge data, we estimate its recharge using an extensive, nationwide database in all 30 hydrological basins of Iran from 2002 to 2017. In this study period, the country's groundwater resources were largely depleted due to anthropogenic and natural drivers. Our results suggest a significant decline in the nationwide groundwater recharge, primarily attributed to unsustainable development practices. By elucidating the anthropogenic and natural drivers contributing to this decline in Iran's groundwater recharge as a representative of a (semi)arid region in the world, we aim to foster a more comprehensive understanding of the regional and global impacts of these large-scale depletions. Our findings advocate for the urgent need for transformative changes in groundwater management and policies, especially in regions experiencing increasing hydroclimatic fluctuations.

## Results and discussion
### Estimated groundwater recharge
The estimated groundwater recharges for each aquifer, totaling 666 aquifers, aggregated in Iran's hydrological basins from 2002 to 2017, are depicted in Fig. 1. The Method's section details how these values were obtained. Our findings reveal a statistically significant decrease in nationwide groundwater recharge of about 35% ($p$ value ≤ 0.001) during the study period, which has resulted in a groundwater imbalance in many basins (Fig. 1). The annual change in groundwater recharge varies from −10.3 mm/yr in the Great Karoon basin to +1.9 mm/yr in the Anzali basin, averaging about −3.8 mm/yr across the country (Table 1). The average groundwater recharge across the country and for the study period is estimated at 39.6 mm/yr, which is slightly above the upper end of the range globally reported for (semi)arid areas (i.e., 0.2 to 35 mm/yr)[15]. However, although Iran is mainly under a (semi)arid climate, some areas are wet with annual mean precipitation exceeding 1000 mm/yr (Fig. S1). As such, the nationwide recharge rate is higher than the global range reported for (semi)arid areas.

The mean amounts of estimated groundwater recharge lie within −44% to +21% of the recharge reported previously for some individual aquifers in Iran[16–20]. Differences between the methods used in the previous studies and the estimation method developed for this study could explain inconsistencies in the results. The observed progressive decline in groundwater recharge could account for the reduction in the studied aquifers in 2022 compared to their previous levels. Moreover, previous studies have also expressed concerns regarding the diminished recharge in various aquifers across the country[21–26]. Overall, the existing literature, alongside the records of Iran's water authorities (see Note S1), unanimously agree that groundwater resource overdrafts and declining recharge have not only pushed the country into a state of water bankruptcy but also posed a significant threat to the country's socioeconomic development.

### Groundwater recharge ratio
The countrywide groundwater recharge ratio (Fig. 2), defined as the fraction of recharge to precipitation, declined from 21% in 2006 to 14%

in 2017, averaging about 17% during our study period, closely approximating the average groundwater recharge ratio at the global scale (i.e., 16%)[3]. The combination of declined recharge and over-exploitation of aquifers has resulted in approximately 422 out of 609 Iranian plains being in a critical/prohibited state. In this context, the term critical/prohibited signifies a condition where digging new wells is prohibited, except for drinking water use. Meanwhile, many of the free plains (i.e., where drilling new wells is legal) are located in desert or mountainous regions, where the aquifers have poor potential for groundwater usage. However, on a smaller geographical scale, i.e., in basins, the average recharge ratio during our study period varied from 3.3% (in Anzali basin) to 39.8% (in Bakhtegan basin) (Fig. 3). In general, the average recharge ratio is lower in the basins with the lowest groundwater depletion such as Anzali and Haraz basins[8], which are located in the wet region in the north of Iran. On the contrary, the average recharge ratio is higher in the basins with the highest decrease in groundwater storage, such as Bakhtegan and Salt Lake basins[27], which are located in relatively dry zones in the central and eastern parts of Iran, where the groundwater resources dominantly contribute to the supply of water for human and environment (up to 90%). Interestingly, both Anzali and Haraz basins, which had the lowest recharge ratio, were the only basins that demonstrated a decreasing trend in groundwater electrical conductivity during the study period[8]. This decline in electrical conductivity could be attributed to the reduced input of contaminants through infiltration into the groundwater[28].

### Main drivers of declined groundwater recharge
While Noori et al.[8] and Ashraf et al.[27] showed that the current extraction rates are unsustainable, other hydroclimatic and anthropogenic drivers (e.g., precipitation, evaporation, and changes in the LUC processes at the surface) governing the decline in aquifer recharge need to be further investigated. To better understand the impact of the natural and anthropogenic drivers on the decline in Iran's groundwater recharge, the changes in the annual mean precipitation across the country and each basin were investigated during the 2000 to 2018 period (Fig. S2). Our results suggest a non-significant change in annual mean precipitation in Iran ($p$ value ≤ 0.05), in line with the results reported by Moshir Panahi et al.[29] for the period from 1986 to 2016. No significant change was also detected in the annual mean precipitation in 26 out of the 30 basins investigated in this study ($p$ value ≤ 0.05). Similarly, no significant change was observed for the evapotranspiration reported in Iran, for the period from 1986 to 2016[29]. Precipitation and (potential) evapotranspiration are the main natural drivers of groundwater recharge. Other natural-based controls on recharge include, among other factors, geology and topography[30]; however, these factors remained relatively stable in Iran over the observation period. This suggests that human interventions have dominantly impacted the decline in Iran's groundwater recharge. It is noteworthy that insignificant changes in natural drivers do not necessarily mean that recharge generation mechanisms remain unchanged. For example, snow contributes more than rainfall to groundwater recharge[31,32]. The study conducted by Safarianzengir et al.[33] highlighted a strong correlation between the decline in groundwater and a decreasing trend in snowfall in Iran (2000–2019). Recent observations also reported a decline in snowfall in different parts of Iran during the last decades[34–36].

However, our results reveal the average amount of nationwide groundwater recharge was 39.6 mm/yr during the study period, which is more than the reported average annual runoff in Iran, i.e., about 32 mm/yr, for the same period (2001–2016)[29], suggesting the national surface water resources are likely seeping into the groundwater basins. This finding is further supported by Saemian et al.[37] and Maghrebi et al.[38] studies, which reported that Iran's surface water resources, as the main contributor of groundwater recharge, have largely declined during the

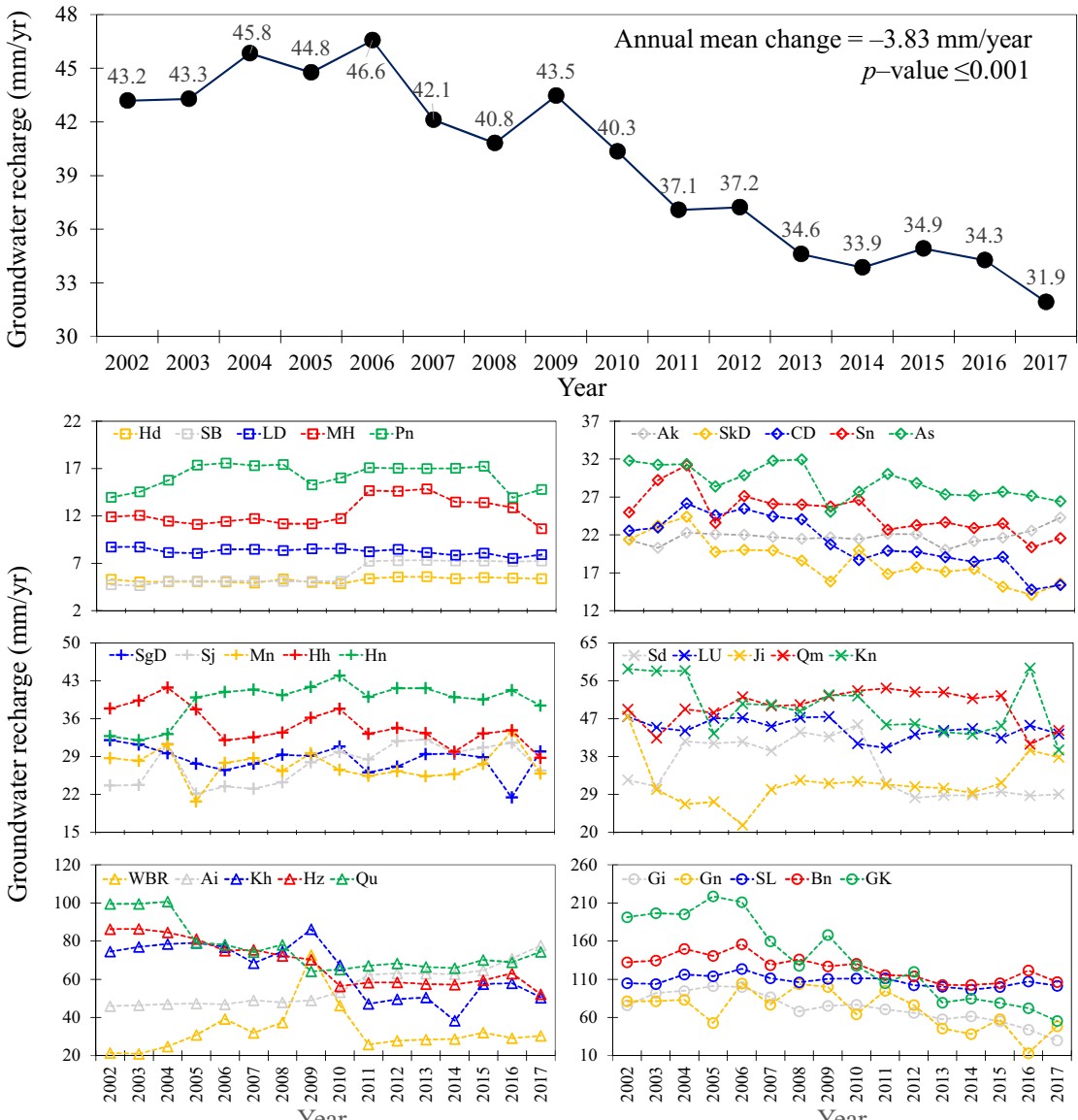

**Fig. 1 | Temporal trend of annual mean groundwater recharge (mm/yr) in Iran (dark color line with circle marker) and in each of the 30 basins from 2002 to 2017.** This figure shows a statistically significant decrease in nationwide groundwater recharge of about 35% ($p$ value $\leq$ 0.001) during the study period. The annual change in groundwater recharge varies from −10.3 mm/yr in the Great Karoon basin to +1.9 mm/yr in the Anzali basin, averaging about −3.8 mm/yr across the country. Hd (light brown color dash line with square marker): Hirmand basin, SB (light gray color dash line with square marker): South Baluchestan basin, LD (blue color dash line with square marker): Lut Desert basin, HM (red color dash line with square marker): Hamun Mashkil basin, Pn (green color dash line with square marker): Patargan basin, Ak (light gray color dash line with diamond marker): Atrak basin, SkD (light brown color dash line with diamond marker): Siahkooh Desert basin, CD (blue color dash line with square marker): Central Desert basin, Sn (red color dash line with diamond marker): Sirjan basin, As (green color dash line with diamond marker): Aras basin, SgD (blue color dash line with plus marker): Saghand Desert basin, Sj (light gray color dash line with plus marker): Sedij basin, Mn (light brown color dash line with plus marker): Mehran basin, Hh (red color dash line with plus marker): Helleh basin, Hn (green color dash line with plus marker): Hamun basin, Sd (light gray color dash line with cross marker): Sefid-roud basin, LU (blue color dash line with cross marker): Lake Urmia basin, Ji (light brown color dash line with cross marker): Jarrahi basin, Qm (red color dash line with cross marker): Qareghom basin, Kn (green color dash line with cross marker): Karian basin, WBR (light brown color dash line with triangle marker): West Boundary River basin, Ai (light gray color dash line with triangle marker): Anzali basin, Kh (blue color dash line with triangle marker): Karkheh basin, Hz (red color dash line with triangle marker): Haraz basin, Qu (green color dash line with triangle marker): Qaresou basin, Gi (light gray color dash line with circle marker): Gavkhouni basin, Gn (light brown color dash line with circle marker): Gorgan basin, SL (blue color dash line with circle marker): Salt Lake basin, Bn (red color dash line with circle marker): Bakhtegan basin, and GK (green color dash line with circle marker): Great Karoon basin.

last decades, mainly due to human interventions. Approximately 56% of Iranian rivers have undergone a decline in stream-flow, which is around 2.5 times higher than values reported for the world's large rivers. Around 20% of the country's permanent rivers have transformed into seasonal rivers, and former seasonal rivers have dried up and/or become narrow streams, leading to the development of new river and fluvial regimes[38]. Anthropogenically dwindling renewable water has also

resulted in shrinkage of almost all of the national lakes, wetlands, marshes, and ponds[8,39]. This, in turn, is likely to have resulted in a reduced aquifer recharge from surface water bodies. A decline in irrigation return flows that can be attributed to the technological progresses of agricultural practices and improvements in the efficiency of irrigation water-use during the last two decades, as suggested by Noori et al.[8], could also contribute to the decline in the country's groundwater

**Table 1 | Trend analysis results and of recharge and the lower and upper limits of the 95% confidence interval in different Iranian basins and across the country**

| Basin | Sen's slope (L) | Significant level (p value) | $L_{min95}$ | $L_{max95}$ |
|---|---|---|---|---|
| Sirjan | −0.485 | ≤0.01 | −0.682 | −0.185 |
| Atrak | 0.027 | >0.05 | −0.075 | 0.164 |
| Aras | −0.334 | ≤0.01 | −0.531 | −0.131 |
| South Baluchestan | 0.188 | ≤0.01 | 0.004 | 0.264 |
| Sedij | 0.558 | ≤0.05 | 0.102 | 0.895 |
| Jarrahi | 0.295 | >0.05 | −0.272 | 0.873 |
| Helleh | −0.496 | ≤0.05 | −0.804 | −0.036 |
| Lake Urmia | −0.183 | >0.05 | −0.459 | 0.057 |
| Salt Lake | −0.804 | ≤0.05 | −1.708 | −0.136 |
| Haraz | −2.423 | ≤0.001 | −2.880 | −1.721 |
| Anzali | 1.931 | ≤0.001 | 1.058 | 2.433 |
| Sefid-roud | −0.455 | >0.05 | −1.414 | 0.083 |
| Bakhtegan | −2.907 | ≤0.01 | −4.349 | −1.466 |
| Gorgan | −3.003 | ≤0.05 | −6.193 | −0.628 |
| Qareghom | 0.252 | >0.05 | −0.324 | 0.604 |
| Great Karoon | −10.297 | ≤0.001 | −13.520 | −8.088 |
| Karkheh | −2.135 | ≤0.01 | −3.513 | −1.039 |
| Mehran | −0.162 | >0.05 | −0.333 | 0.209 |
| Saghand Desert | −0.115 | >0.05 | −0.485 | 0.127 |
| Siahkooh Desert | −0.510 | ≤0.001 | −0.699 | −0.343 |
| Lut Desert | −0.051 | ≤0.01 | −0.077 | −0.018 |
| Central Desert | −0.660 | ≤0.001 | −0.894 | −0.384 |
| Gavkhouni | −3.955 | ≤0.001 | −5.148 | −2.768 |
| West Boundary River | 0.490 | >0.05 | −0.459 | 1.165 |
| Karian | −0.99 | ≤0.05 | −1.360 | −0.224 |
| Patargan | −0.01 | >0.05 | −0.082 | 0.223 |
| Hamun | 0.23 | >0.05 | −0.111 | 0.745 |
| Hamun Mashkil | 0.10 | >0.05 | −0.074 | 0.266 |
| Hirmand | 0.03 | >0.05 | −0.004 | 0.054 |
| Qaresou | −1.56 | ≤0.05 | −2.957 | −0.354 |
| Iran | −3.83 | ≤0.001 | −4.240 | −2.682 |

$L_{min95}$ and $L_{max95}$ are the lower and upper limits of the 95% confidence interval of calculated Sen's slope (L), respectively.

recharge—a concept known as pendulum swing[40]. Maghrebi et al.[41] proposed that one of the key factors contributing to the failure of government plans for aquifers restoration through modernization of the irrigation systems was the decline in groundwater recharge, primarily caused by reductions in irrigation return flows. This reduction in recharge led to a significant increase in the nationwide groundwater deficit, escalating from 106 km³ to more than 140 km³ during a 12-year period. Intensive and progressive land subsidence across the country during the last two decades has also contributed to reduced soil permeability due to compaction, thereby impacting the capacity for groundwater recharge[42–45]. In support of an effect of land subsidence, Iran is ranked among the countries with highest rate of land subsidence globally[25]. As a result, aquifers' capacities in Iran are declining due to countrywide land subsidence. In this scenario, the soil compaction reduces its permeability and hence the aquifers' ability to be recharged, leading to a decline in both the rate and the total volume of recharge. Another factor influencing groundwater recharge is the changes in LUC, as several regions in Iran have experienced extensive deforestation, desertification, and rapid urbanization[46–51].

## Implications of reduced groundwater recharge

Reduction in groundwater recharge combined with the nonrenewable groundwater discharge from the country's aquifers[8,27,52] further contributes to decline in groundwater storage and shrink in groundwater table. The current declining trend in groundwater recharge and the reducing trend in groundwater table reported by Noori et al.[8] and Ashraf et al.[27] persist through this century under status of shrinking renewable water and the quo water and environmental resources management in Iran. It has resulted in a gradual exacerbation of Iran's water and food security and makes the country's landscapes prone to a wider and more severe range of already experienced disasters, such as desertification, dust storms, landslides, land subsidence, sinkholes, droughts, floods, and fires. Additionally, it leads to a decrease in soil fertility, biodiversity, and an increase in greenhouse gas emission. From a regional perspective, such a decline in groundwater recharge has devastating consequences for the country's exports of agricultural products to the neighboring countries, compromising regional food security, given that Iran is one of the major food producers in the Persian Gulf region.

Artificial recharge is undoubtedly a promising engineering solution to recover depleted groundwater resources globally. Iran is also making efforts to complete a national plan that aims to artificially recharge groundwater up to about 1 km³. However, it should be noted that this amount is considerably less than the deficit in the country's

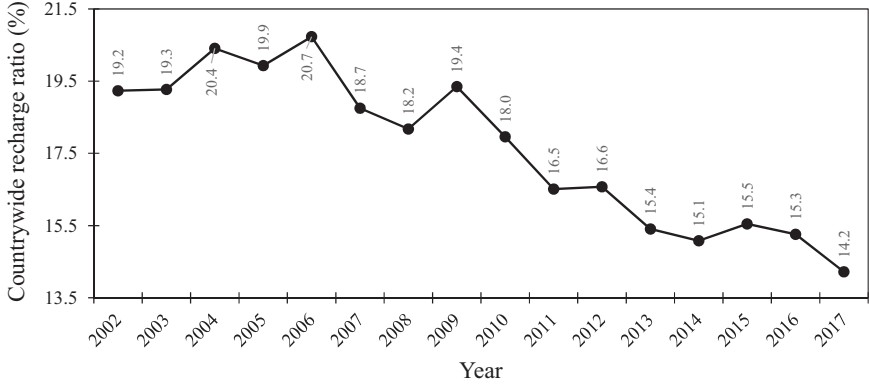

**Fig. 2 | Temporal trend of groundwater recharge ratio (%), i.e., ratio of recharge to precipitation, in Iran from 2002 to 2017.** The countrywide groundwater recharge ratio declined from 21% in 2006 to 14% in 2017, averaging -17% during the study period, closely approximating the average groundwater recharge ratio at the global scale (i.e., 16%).

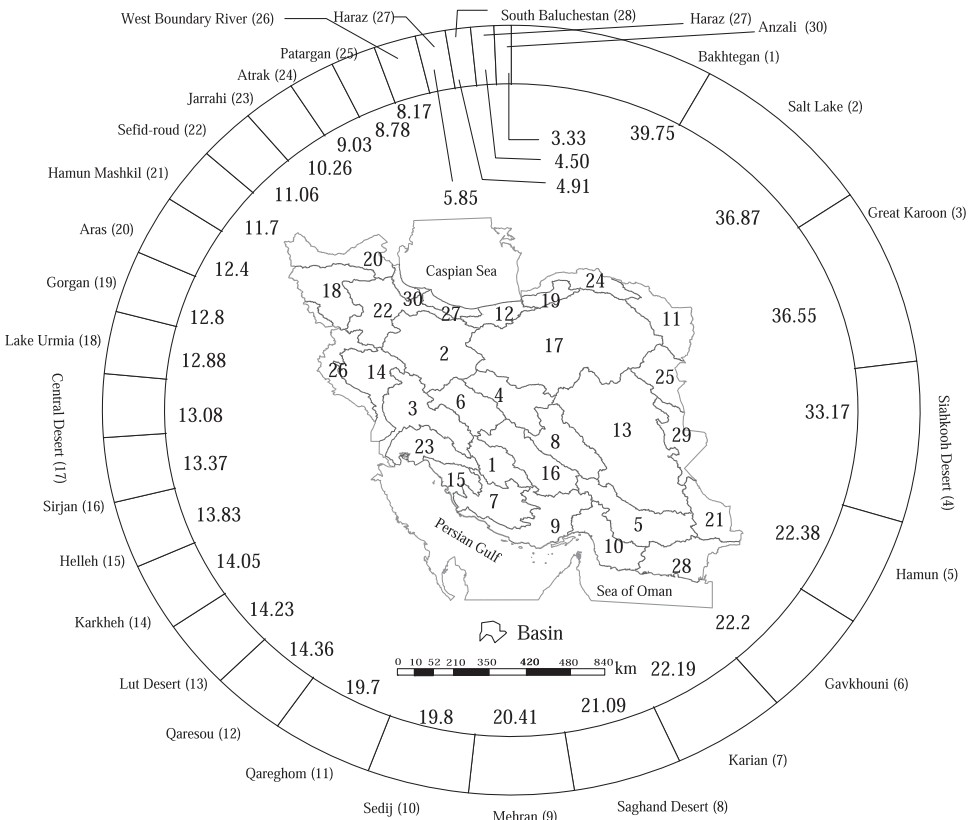

**Fig. 3 | Mean groundwater recharge ratio (%) in each Iranian basin from 2002 to 2017.** The average recharge ratio during the study period varied from 3.3% (in Anzali basin) to 39.8% (in Bakhtegan basin).

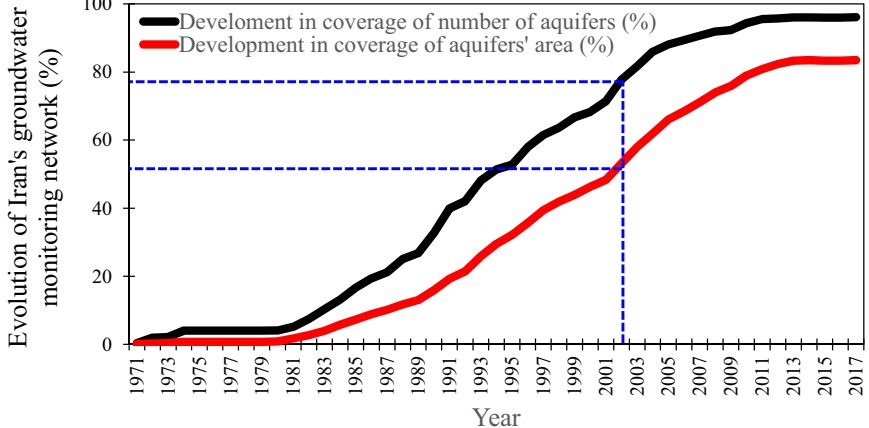

**Fig. 4 | Evolution of groundwater monitoring network in Iran.** The blue line shows the onset of our study when the monitoring network covers around 78% of the national aquifers' area and >50% of the number of aquifers in Iran.

groundwater resources (see Note S1). Iran's water resource management experiences highlight that engineering solutions alone cannot prevent the alarming depletion of groundwater resources for the country[41], particularly when other factors such as countrywide land subsidence can hinder the effectiveness of recharge efforts. Our primary suggestion for facilitating the restoration of the depleted aquifers is prioritizing and reducing the water demand instead of relying on nonrenewable groundwater use. This recommendation requires adequate water governance backed by efficient conservation policies that are not currently implemented in Iran. The poor governance of hydro-environmental resources, characterized by unsustainable LUC planning, inequitable allocation rules, institutional ineffectiveness,

and poor economic conditions, makes it challenging to rehabilitate groundwater resources through reduction in abstraction and implementation of efficient artificial recharge strategies. To address this concern, the existing top-down governance structure approach should be replaced with a cooperative and bottom-up approach that considers the interests of local stakeholders, particularly farmers, as the main consumers of groundwater resources.

## Methods
### Study area and data
Iran's groundwater monitoring network has been gradually established since the early 1970s (Fig. 4 and Note S2). The network covers 492 out

of 609 plains (Fig. S3), including 666 aquifers (Fig. S4), and contains 12,293 active piezometers (Fig. S4) that measure the groundwater level at monthly intervals. The total area of aquifers is around 272,000 km$^2$ and the monitoring network covered more than 96% of this area (i.e., 261,000 km$^2$) by the end of 2017. The remaining area (i.e., ~11,000 km$^2$), includes 133 relatively small and less important aquifers distributed across the country[5].

Although the groundwater levels were measured in active piezometers from the early 1960s, the comprehensive database of Iran's groundwater resources was established from the water year of 2001–2002, when ~78% of the aquifer areas became covered by the groundwater monitoring network[53]. However, this database does not include groundwater recharge. To address this knowledge gap, we utilized the national groundwater database to calculate the groundwater recharge across Iran in each of the 30 basins during 2002 to 2017 period. This time interval was selected as our study period due to the data availability and adequate coverage of the country's aquifers using the groundwater monitoring network (Note S2).

### Groundwater recharge calculation
Groundwater recharge includes any penetrated water into the saturated zone[54]. We used the water budget method, i.e., Eq. (1)[15], to estimate groundwater recharge in each aquifer across Iran.

$$\text{Recharge} = Q_{off} + \triangle S \qquad (1)$$

where, $Q_{off}$ equals egressed water from the aquifer (positive flux out of aquifer), and $\triangle S$ is the groundwater storage change (negative when storage decreases) (see, Fig. S5).

In this study, the representative hydrograph was determined for each aquifer. Then the annual change in groundwater storage was estimated by considering the aquifer area, specific yield, and the annual average change in the groundwater level. Given the large number of aquifers investigated in this study (i.e., 666), we aggregated the calculated recharge to obtain values for each hydrological basin and for the entire country. Details about the aggregated data are given in Note S3 and also supplemented in https://zenodo.org/record/8382150.

Egressed water from the aquifers (i.e., $Q_{off}$) includes all anthropogenic groundwater withdrawals and natural groundwater losses. Anthropogenic groundwater withdrawals are fully measured based on sampling campaigns and field inventories in over a million wells and qanats on approximately a five-year cycle during wet and dry periods in Iran. The same measurements as the 5-year field campaigns are done twice a year during wet and dry periods at a small but representative set of selected extraction points in different aquifers. Natural groundwater losses can take place through different routes such as natural springs, evapotranspiration, and outgoing lateral fluxes from the aquifers. Fortunately, the national groundwater database includes the annual groundwater discharge through natural springs, which is measured as frequently as the anthropogenic groundwater withdrawals and thus also based on sampling campaigns and field inventories. Evaporation through the saturated zone is usually negligible when the depth to the groundwater table is large (>3 m depth). We supposed that the groundwater loss through evaporation, including transpiration, is negligible because the groundwater depth was more than 3 m in almost all the aquifers[8]. Groundwater losses through aquifer discharge to surface waters are also negligible due to the severe decline in the countrywide groundwater table during the last five decades (>20 m)[55]. This hypothesis gains further support when data show both a widespread decline in Iran's river flows[38] and drying out the national wetlands and inland lakes[8,56]. The natural groundwater losses through outgoing lateral groundwater fluxes are not considered due to the lack of data, which introduces uncertainty to the recharge estimate for individual aquifers. However, for the

nationwide recharge estimate, these losses can be reasonably ignored by disregarding cross-border groundwater fluxes. It should be noted that incoming lateral fluxes from the adjacent aquifers are represented in the recharge itself. Another issue is that surface water basins are determined by topography, and they do not necessarily match with regional aquifers, for which the territory is identified by underground geology. As a result, depending on how aquifers and surface drainage basins are situated in a given region, there might be additional fluxes, both incoming and outgoing to a given aquifer, which represent how the aquifer interacts with adjacent drainage basins. Lack of data in this regard limited our ability to consider such detailed information in our study. However, only 6 out of 666 studied aquifers were shared in two adjacent drainage basins. Therefore, we expected that this issue would have a minimal impact on our results. In summary, the simplifications implemented in this study may introduce uncertainties in the results. To address this issue, we conducted an uncertainty analysis that accounts for the impacts of such simplifications in our study.

### Data analysis
Groundwater data processing, including aggregation, division, multiplication, normalization, and statistical analysis, for 666 national aquifers during the study period (2002–2017), was performed using Pivot Table in Microsoft Excel environment, ArcGIS, and MATLAB. All figures were drawn in ArcGIS and MATLAB environments, Microsoft Office software, and Grapher program.

For a description of the change in groundwater recharge and precipitation, the magnitude of temporal trends was calculated by Sen's slope estimator method[57]. A Mann–Kendall test[58,59] was also applied to differentiate between significant and insignificant trends in both groundwater recharge and precipitation. These statistical analyses were done by MAKESENS 1.0 software, introduced by the Finnish Meteorological Institute[60].

### Data uncertainty
The results obtained in this study heavily rely on the data quality of the national groundwater database. Factors including poor information on aquifer type (e.g., confined or unconfined aquifer types), specific yield, and hypotheses made can potentially introduce errors/uncertainties in the calculated groundwater recharge for Iran and in each of the 30 basins. Detailed information on the data and the corresponding sources of errors/uncertainties is given by Noori et al.[8].

An uncertainty analysis was conducted by determining the lower and upper limits of the 95% confidence interval for magnitude of the trends in groundwater recharge calculated using Sen's slope estimator[53] (see the Methods section for details). Table 1 shows the results of the uncertainty analysis for the calculated groundwater recharge across 30 basins and the countrywide values. By considering a significant level of 0.05 (i.e., $\alpha = 0.05$), the lower and upper limits of the confidence interval show the acceptable deviation from the magnitude of the trends for groundwater recharge across different basins and for the entire country.

## Data availability
The raw data used in this study are publicly available via Data Archive of the Iran Water Resources Management Company (IWRMC) at https://stu.wrm.ir/register.asp. Alternatively, the processed data presented in this paper are supplemented in https://zenodo.org/record/8382150.

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

## Acknowledgements

We thank the Iran Water Resources Management Company for supporting us with groundwater data.

## Author contributions

R.N. and M.M. conceived the study. R.N. and M.M. carried out the analyses. R.N., M.M., S.Je., S.M.B., S.Ja. and M.N. contributed to the data. R.N., S.Je., S.M.B., E.H., S.P., S.A. and A.A. contributed discussions and modeling insights. R.N. wrote the article. All authors reviewed, edited, and approved the manuscript.

## Competing interests

The authors declare no competing interests.
