## [Peer Review File · Nature Communications]

Decline in Iran's groundwater rechargeREVIEWER COMMENTS

Reviewer #1 (Remarks to the Author):

Review of "Decline in Iran's Groundwater Recharge" by Noori et al.

This is an interesting study that shows how not only the groundwater levels have been declining in Iran's aquifer, but also that groundwater recharge has been dwindling. This is fact would provide a double-jeopardy for Iran's water resources and the economy and wellbeing that depends on it.

I think the study, given that is based on a wealth of data, and Iran being an important depletion hotspot, is worth being published in Nature Communications after some consideration.

1. The way the declining recharge is coupled to groundwater development as framed as a vicious cycle is flawed. The reduced discharge is likely separate from this, as the authors show: it likely comes from declined streamflow infiltration (concentrated recharge), reduced return flows (which may for instance be caused by more efficient irrigation technology) and increased potential evapotranspiration (climate change). This should be made more clear in the abstract while the term vicious cycle avoided later on since it suggests a positive feedback between groundwater decline and reduced recharge, while one expect a negative feedback in fact: lower groundwater tables reduce soil wetness and therefore enhance groundwater recharge.
2. The causes behind declined recharge should be further explored. 1) How is streamflow changing over time (there must be measurements), what are the causes and are these declining streams also the ones that are infiltrating with respect to groundwater levels? 2) Have irrigation technologies changed over the period, i.e. become more efficient? 3) What happened to temperature and therefore potential evapotranspiration (atmospheric demand) over time? This would shed light on which of these causes is most important and where.
3. The data underlying the methods are quite limitedly displayed in the paper. I understand the raw data are available under request, but the aggregated results for each aquifer could be provided in the SI. Thus, for each aquifer it would be good to show figures with in each figure the time series of Q_{pumping} , $Q_{\text{discharge_from_springs}}$, ΔS and finally the resulting Recharge with a regression line fitted through it (based on the equation in the Methods). Also, these time series data should be made available through some repository. Finally, the coordinates of wells and springs and their characteristics (including average groundwater level and Sen's slope) could be presented in a large table in the Supplementary. This would make the paper and methods as FAIR as possible.
4. There is a discussion on uncertainties, but no uncertainty analysis is presented. It must be possible using a simple uncertainty analysis (assuming statistical independence) based on uncertainties in aquifer average Q_{pumping} , $Q_{\text{discharge_from_springs}}$, ΔS to obtain uncertainty estimates of Recharge. Or, if trends as estimated the standard deviation of the Sen's slope reported.

Reviewer #2 (Remarks to the Author):

This manuscript evaluates the decline in groundwater recharge in the Iranian basin from 2002 to 2017 by use of the Iran groundwater monitoring network. Overall, the authors provided novel and important results for the public, scientific community, and policymakers. The manuscript is written well and it is in good shape, and I recommend acceptance of this paper in Nature Communications, after addressing the concerns listed below.

1. In the estimation of groundwater recharge, various sources of uncertainty accumulate such as surface water and groundwater connections, evapotranspiration, and local geology and land use. The authors addressed surface water and evapotranspiration in the method section. I would like to ask the authors' justifications about local geology and land use in groundwater recharge estimation.
2. The estimation of groundwater recharge, could be varied based on the climate of the study area. For example, the groundwater recharge rate has more important in arid and semi-arid

regions due to limited precipitation and their susceptibility to drought. So, it is recommended to distinguish the results for the estimation of groundwater recharge in Iran's basin based on the different climates in Iran.

3. Based on a previous study conducted by Ashraf et al. 2021, excessive groundwater withdrawal in Iran has leading in decreases soil stability in Iran basins, would you please justify how this change can affect groundwater recharge in the study area?

4. What are the authors' suggestions to policymakers and water managers in Iran to take measures in advance to improve groundwater recharge in Iran?

Reference: Ashraf, S., Nazemi, A. & AghaKouchak, A. Anthropogenic drought dominates groundwater depletion in Iran. *Sci Rep* 11, 9135 (2021). <https://doi.org/10.1038/s41598-021-88522-y>

Reviewer #3 (Remarks to the Author):

"Decline in Iran's Groundwater Recharge" tries to provide a new understanding about the massive groundwater depletion in Iran, this time from the lens of declining recharge. The issue of regional groundwater depletion in the MENA region in general and Iran in particular is a known crisis since the GRACE data have become available. As moved forward, GRACE findings have been validated through in-situ information. Restating and reframing this issue from the new perspective of recharge that can result in a hydrological explanation of this phenomenon, therefore, are both relevant and timely and can provide a better insight about the nature of groundwater depletion in Iran and globally.

I have a couple of points for authors' considerations and response for the next round of review:

1) Equation 1 ($Recharge = Q_{off} \pm \Delta S$) presents the core conceptual/mathematical framework with which the recharge is calculated across Iran's basins based on the changes in aquifer storage and outgoing fluxes. I have a couple of concerns here. First, I am not able to understand the reason for \pm sign for changes in the storage. If Q_{off} is an outgoing flux and recharge is an incoming flux then we should have only + sign for changes in the storage. Please clarify. In addition, second, it seems that we have a more severe issue as there might be some other terms in the aquifer balance equation that are not seen here. Most importantly, the groundwater/surface water interaction and the incoming and outgoing fluxes from one aquifer to adjacent aquifers are not represented in this equation. Authors mentioned in the text that the Q_{off} consider both human and natural outfluxes from the aquifers, but what about incoming fluxes beyond basin recharge, for instance as return flow from irrigated lands? I can understand that this may be represented in the recharge itself, but the problem is still there. The key issue is that surface water basins are determined by topography and they do not necessarily match with regional aquifers, for which the territory is identified by underground geology. As a result depending on how aquifers and surface drainage basins are situated in a given region, there might be additional fluxes, both incoming and outgoing to a given aquifer, which represent how the aquifer is interacting with adjacent aquifers and/or drainage basins. It should be clarified how these terms are seen here, and if they are not seen what the implications of this major simplifications would be.

2) Authors noted to a number of works that have used in-situ data to estimate groundwater depletion in Iran. There are even more studies that use satellite estimates and indirectly estimated the recharge. Is the estimates of recharge coming from this work confirming previous estimates of recharge? If the estimates are different, it should be clarified to what extent, in which temporal and spatial scales, and more importantly why. I believe the value of this work will be much more if somehow the estimates made in here are positioned with respect to previous studies.

3) Without more information, I would be a bit sceptic about the conclusion made that the decline in the recharge is solely due to unsustainable management of water resources. There is no doubt that unsustainable management of water resources have a significant role in here, but it does not mean that if the course of unsustainable actions are reversed today, then the natural recharge

would be replenished. Numerous reports on Iran's land subsidence point to the fact that aquifers' capacities in Iran (not only their storages) are declining too. In that case, due to the soil compaction, infiltrability and accordingly the rate and the total volume of recharge declines too. Therefore, even if water management decisions are changed, still the recharge cannot be recovered, because past mismanagements affected the natural capacity of recharge today (and perhaps forever). Another issue is that recharge is not only affected by the decision related to water resource management per say. Activities related to land use and broader environmental management, e.g. urbanization, deforestation, etc., can also affect the rate of recharge tremendously. I found the argument of the authors rather unconvincing that vegetation and land cover are rather unchanged during the study period, as Iran are among nations with fastest rates of land cover changes due to deforestation, desertification and aggressive urbanization. As a result, the effect of vegetation, and land use/management should not be neglected. Also even if the amount of P and E have insignificant changes, it does not necessary means that runoff/recharge generation mechanisms remain unchanged. For instance form of P (snow vs. rain) can be important for recharge. The issue of vegetation and changes in the infiltrability, which are also noted above, are two other evidence showing that insignificant changes in P and E are not strong evidences for insignificant changes in natural mechanisms that generate recharge.

RESPONSE LETTER TO COMMENTS ON “Decline in Iran’s Groundwater Recharge”
MANUSCRIPT NUMBER: NCOMMS-23-07744.R1

Response to Comments of Reviewer #1

General Comment

- This is an interesting study that shows how not only the groundwater levels have been declining in Iran’s aquifer, but also that groundwater recharge has been dwindling. This is fact would provide a double-jeopardy for Iran’s water resources and the economy and wellbeing that depends on it. I think the study, given that is based on a wealth of data, and Iran being an important depletion hotspot, is worth being published in Nature Communications after some consideration.
- **Response**
We would like to thank Reviewer #1 for their positive evaluation of our work and their valuable feedback. We have carefully considered your feedback and revised our manuscript accordingly to address all your comments.

Comment #1

- The way the declining recharge is coupled to groundwater development as framed as a vicious cycle is flawed. The reduced discharge is likely separate from this, as the authors show: it likely comes from declined streamflow infiltration (concentrated recharge), reduced return flows (which may for instance be caused by more efficient irrigation technology) and increased potential evapotranspiration (climate change). This should be made more clear in the abstract while the term vicious cycle avoided later on since it suggests a positive feedback between groundwater decline and reduced recharge, while one expect a negative feedback in fact: lower groundwater tables reduce soil wetness and therefore enhance groundwater recharge.

- **Response**

Thank you for your comment.

In general, we agree that it may exist negative feedback between groundwater decline and reduced recharge: lower groundwater tables reduce soil moisture and therefore enhance groundwater recharge. This feedback is known to be strong in relatively wet environments (e.g., most of Europe). However, most of our study area is in semi-arid, arid and hyper arid environments. In such environments, this feedback is not very strong. For example, groundwater level in many wells in the south of Fars, Kerman, Esfahan, Semnan, Hamedan, Razavi Khorasan, and Tehran provinces have dropped down from around 10s to 100s of meters during the last two decades. Now, some drinking wells in Fars province are even more than 500 m deep. This changes are very significant, but they do not influence the moisture level at the surface. Hence, this specific feedback is very week.

Furthermore, reduction in soil moisture does not necessarily enhance groundwater recharge. If there is available water for recharge, lower moisture levels might increase recharge. However, that is not the case in our study area. Simply, there is no water left for recharge. Long-term monitoring data shows that, approximately 56% of the Iranian rivers have undergone a decline in streamflow, with the decline being around 2.5 times higher than that reported for the world’s large rivers. Around 20% of the country’s permanent rivers have transformed to seasonal rivers and former seasonal rivers have dried up and/or become narrow streams. Anthropogenically dwindling renewable

water has also resulted in shrinkage of almost all national lakes, wetlands, marshes, and ponds, with some of these water bodies passed the point of no return and undergoing complete annihilation. In addition, groundwater depletion has resulted in intensive and a progressive land subsidence across Iran during the past two decades. Iran is ranked among the countries with highest rate of land subsidence globally. As such, aquifers' capacities in Iran (not only their storages) are also declining, primarily due to the countrywide land subsidence. Due to soil compaction, the infiltrability decreases, which subsequently leads to a decline in the rate and total volume of recharge (Lines 148-199 and Text S1).

Comment #2

- The causes behind declined recharge should be further explored. 1) How is streamflow changing over time (there must be measurements), what are the causes and are these declining streams also the ones that are infiltrating with respect to groundwater levels? 2) Have irrigation technologies changed over the period, i.e. become more efficient? 3) What happened to temperature and therefore potential evapotranspiration (atmospheric demand) over time? This would shed light on which of these causes is most important and where.

- **Response**

Thank you for your comment.

We have now made a revised and enhanced discussion on the underlying cause and processes behind the declined recharge (Lines 148-199).

1) Our revised manuscript provides additional discussion on the countrywide declining streamflow, and surface water resources, and elaborated on the main underlying causes of this in Iran by referring to the recent studies conducted by Saemian et al. (2022) and Maghrebi et al. (2023) (Lines 172-183).

2) Yes. To relieve Iran's water crisis, the government invested about US\$1.5 billion in modernizing the country's irrigation systems, aiming to restore groundwater resources. At the start of the program (in 2011), Iran's groundwater storage had fallen by about 106 billion cubic meters since 1965. But the government investment and modernization of irrigation systems did not result in improving this deficit. By 2022, field measurements indicated that groundwater deficit had increased to about 140 billion cubic meters. However, the increased competition for the consumption of the water saved by the program resulted in intensified agricultural development and even higher water demand. The failure of the program can be attributed to the development of bare lands, reductions in groundwater recharging by returning flows from irrigation, and changes in cultivation patterns across the country. The manuscript has now revised to reflect these issues in our discussion (Lines 183-191).

3) You are correct about the possible effects of climatic conditions on the recharge. The manuscript has elaborated on this issue in depth, and we discussed the changes in evapotranspiration and even in the form of precipitation in Iran by referring to the recent conducted studies (Lines 148-171).

Comment #3

- The data underlying the methods are quite limitedly displayed in the paper. I understand the raw data are available under request, but the aggregated results for each aquifer could be provided in the SI. Thus, for each aquifer it would be good to show figures with in each figure the time series of $Q_{pumping}$, $Q_{discharge_from_springs}$, ΔS and finally the resulting Recharge with a regression line fitted through it (based on the equation in the Methods). Also, these time series data should be made available through some repository. Finally, the coordinates of wells and springs and their characteristics (including average

groundwater level and Sen's slope) could be presented in a large table in the Supplementary. This would make the paper and methods as FAIR as possible.

- **Response**

The raw data used in this study are publicly available via Data Archive of the Iran Water Resources Management Company (IWRMC) at <https://stu.wrm.ir/register.asp>. Alternatively, the data presented in this paper are now supplemented in <https://zenodo.org/record/8179231>

Unfortunately, we did not have access to the coordinates of springs, Qanats, and wells. Thus, we could not present spatial distribution of these extraction points in Iran (more than one million points).

Comment #4

- There is a discussion on uncertainties, but no uncertainty analysis is presented. It must be possible using a simple uncertainty analysis (assuming statistical independence) based on uncertainties in aquifer average $Q_{pumping}$, $Q_{discharge_from_springs}$, ΔS to obtain uncertainty estimates of Recharge. Or, if trends as estimated the standard deviation of the Sen's slope reported.

- **Response**

Thank you for your constructive comment.

We have conducted an uncertainty analysis and introduced this to the manuscript (Lines 234-250 and Table 1).

Response to Comments of Reviewer #2

General Comment:

- This manuscript evaluates the decline in groundwater recharge in the Iranian basin from 2002 to 2017 by use of the Iran groundwater monitoring network. Overall, the authors provided novel and important results for the public, scientific community, and policymakers. The manuscript is written well and it is in good shape, and I recommend acceptance of this paper in Nature Communications, after addressing the concerns listed below.
- **Response**
We would like to thank Reviewer #2 for their positive evaluation of our work and the constructive feedback that helped us to shape our paper well. We have carefully considered your helpful feedback and revised our manuscript accordingly.

Comment #1

- In the estimation of groundwater recharge, various sources of uncertainty accumulate such as surface water and groundwater connections, evapotranspiration, and local geology and land use. The authors addressed surface water and evapotranspiration in the method section. I would like to ask the authors' justifications about local geology and land use in groundwater recharge estimation.
- **Response**
Thank you for your comment.
Local geology can be considered as a rather stable control over the observation period (Lines 162-165). However, land use/cover was subject to continual changes during the study period. We added new discussion in the revised manuscript to elaborate on the influences from land use/cover (Lines 197-199). In addition, we added further discussion about the other significant drivers of declined recharge, relevant to the case

of Iran, e.g. land subsidence (Lines 191-197) and changes in the form of precipitation, i.e., declined snowfall (Lines 166-171).

Comment #2

- The estimation of groundwater recharge, could be varied based on the climate of the study area. For example, the groundwater recharge rate has more important in arid and semi-arid regions due to limited precipitation and their susceptibility to drought. So, it is recommended to distinguish the results for the estimation of groundwater recharge in Iran's basin based on the different climates in Iran.

- **Response**

Thank you for your comment.

Although our method proposed for estimation of recharge is applicable for all climate conditions, the results presented in this study, in agreement with the previous studies, show a spatial variation based on the aridity of specific regions. The average groundwater recharge across the country and for the entire study period (2002-2017) was estimated at 39.6 mm/yr, which is just above the upper end of the range reported for (semi)arid regions in the world (i.e., 0.2 to 35 mm/yr) by Scanlon et al. (2002). However, although Iran is mainly covered by (semi)arid lands, some areas are wet with annual mean precipitation exceeding 1000 mm/yr. Therefore, the nationwide recharge rate is slightly higher than the range reported for (semi)arid regions in the globe (Lines 98-104). In addition, we have now added further discussion to the revised manuscript, to elaborate on and distinguishing the difference between the estimated recharge rates in dry and wet regions of Iran (Lines 133-138).

Comment #3

- Based on a previous study conducted by Ashraf et al. 2021, excessive groundwater withdrawal in Iran has leading in decreases soil stability in Iran basins, would you please justify how this change can affect groundwater recharge in the study area?

- **Response**

Thank you for bringing this point to our attention.

We appreciate the suggested useful paper, which can certainly help us to further justify and sense check our results. Soil stability, sinkholes, landslides, and land subsidence can affect groundwater recharge. We have now added a full discussion about the change in the soil form (mainly land subsidence) and its impacts on the declined groundwater recharge in Iran (Lines 191-199, 220-223, Text S1).

Comment #4

- What are the authors' suggestions to policymakers and water managers in Iran to take measures in advance to improve groundwater recharge in Iran?

- **Response**

We have now added a new paragraph to the manuscript, which suggest possible measures to policymakers for enhancing groundwater recharge (Lines 217-233).

Response to Comments of Reviewer #3

General Comment

- “Decline in Iran’s Groundwater Recharge” tries to provide a new understanding about the massive groundwater depletion in Iran, this time from the lens of declining recharge. The issue of regional groundwater depletion in the MENA region in general and Iran in

particular is a known crisis since the GRACE data have become available. As moved forward, GRACE findings have been validated through in-situ information. Restating and reframing this issue from the new perspective of recharge that can result in a hydrological explanation of this phenomenon, therefore, are both relevant and timely and can provide a better insight about the nature of groundwater depletion in Iran and globally. I have a couple of points for authors' considerations and response for the next round of review:

- **Response**

We would like to thank Reviewer #3 for their positive evaluation of our work and their constructive feedback that helped us to better shape our paper. We have carefully considered your helpful feedback and revised our manuscript accordingly.

Comment #1

- Equation 1 ($Recharge = Q_{off} \pm \Delta S$) presents the core conceptual/mathematical framework with which the recharge is calculated across Iran's basins based on the changes in aquifer storage and outgoing fluxes. I have a couple of concerns here. First, I am not able to understand the reason for \pm sign for changes in the storage. If Q_{off} is an outgoing flux and recharge is an incoming flux then we should have only $+$ sign for changes in the storage. Please clarify. In addition, second, it seems that we have a more severe issue as there might be some other terms in the aquifer balance equation that are not seen here. Most importantly, the groundwater/surface water interaction and the incoming and outgoing fluxes from one aquifer to adjacent aquifers are not represented in this equation. Authors mentioned in the text that the Q_{off} consider both human and natural outfluxes from the aquifers, but what about incoming fluxes beyond basin recharge, for instance as return flow from irrigated lands? I can understand that this may be represented in the recharge itself, but the problem is still there. The key issue is that surface water basins are determined by topography and they do not necessarily match with regional aquifers, for which the territory is identified by underground geology. As a result depending on how aquifers and surface drainage basins are situated in a given region, there might be additional fluxes, both incoming and outgoing to a given aquifer, which represent how the aquifer is interacting with adjacent aquifers and/or drainage basins. It should be clarified how these terms are seen here, and if they are not seen what the implications of this major simplifications would be.

- **Response**

Thank you for your comment.

We agree with the point that you raised and have revised Equation (1) accordingly (Line 274).

As you correctly mentioned, the groundwater/surface water interaction, the incoming fluxes from one aquifer to adjacent aquifers, and the incoming fluxes beyond basin recharge (e.g., return flow from irrigated lands) are be represented in the recharge itself. We also agree with you regarding the natural difference between the hydrological basins and aquifers. Unfortunately, lack of data made it impossible to consider such detailed information in our study. However, it is worth noting that only 6 out of 667 studied aquifers were shared between two adjacent basins. Therefore, this issue have a minimal impact on the results presented in this study. We have now clarified this issue in the text of the revised manuscript. In addition, we carried out an uncertainty analysis that would somewhat cover such simplifications in our study (Lines 234-250, 279-282, 289-290, 299-313, Table 1, Text S3, and Figure S5).

Comment #2

- Authors noted to a number of works that have used in-situ data to estimate groundwater depletion in Iran. There are even more studies that use satellite estimates and indirectly estimated the recharge. Is the estimates of recharge coming from this work confirming previous estimates of recharge? If the estimates are different, it should be clarified to what extent, in which temporal and spatial scales, and more importantly why. I believe the value of this work will be much more if somehow the estimates made in here are positioned with respect to previous studies.
- **Response**
Thank you for your comment.
We have compared our findings with the results of relevant studies conducted in some individual aquifers in Iran. In addition, we discussed the underlying reasons for the possible inconsistencies that may exist between our results and those reported by previous studies (Lines 105-119 and Text S1).

Comment #3

- Without more information, I would be a bit sceptic about the conclusion made that the decline in the recharge is solely due to unsustainable management of water resources. There is no doubt that unsustainable management of water resources have a significant role in here, but it does not mean that if the course of unsustainable actions are reversed today, then the natural recharge would be replenished. Numerous reports on Iran's land subsidence point to the fact that aquifers' capacities in Iran (not only their storages) are declining too. In that case, due to the soil compaction, infiltrability and accordingly the rate and the total volume of recharge declines too. Therefore, even if water management decisions are changed, still the recharge cannot be recovered, because past mismanagements affected the natural capacity of recharge today (and perhaps forever). Another issue is that recharge is not only affected by the decision related to water resource management per say. Activities related to land use and broader environmental management, e.g. urbanization, deforestation, etc., can also affect the rate of recharge tremendously. I found the argument of the authors rather unconvincing that vegetation and land cover are rather unchanged during the study period, as Iran are among nations with fastest rates of land cover changes due to deforestation, desertification and aggressive urbanization. As a result, the effect of vegetation, and land use/management should not be neglected. Also, even if the amount of P and E have insignificant changes, it does not necessary means that runoff/recharge generation mechanisms remain unchanged. For instance, form of P (snow vs. rain) can be important for recharge. The issue of vegetation and changes in the infiltrability, which are also noted above, are two other evidence showing that insignificant changes in P and E are not strong evidences for insignificant changes in natural mechanisms that generate recharge.
- **Response**
Thank you for your constructive comment.
We agree with your comment. Our statements could reflect the sense that the decline in the recharge is solely due to unsustainable management of water resources. Given your insightful suggestions, we have now made a full discussion on the results and concluded that “This decline is primarily attributed to unsustainable water and environmental resources management, exacerbated by decadal changes in climatic conditions. However, it is important to note that the former’s contribution outweighs the latter” (Lines 28-31). We have now revise the manuscript to avoid this ambiguity for the readers in different parts of the manuscript (Lines 28-31, 148-199, 220-223, Text S1).

We agree that land uses/covers were changing during the study period. In addition, you are correct about the impact of land subsidence on the decline of aquifers' capacities in Iran. We have now revised the manuscript to address these issues in the manuscript (Lines 191-199, 220-223, Text S1).

As you correctly mentioned, the form of precipitation can affect the groundwater recharge. We have now discussed about this issue and enhanced our discussion with some relevant studies that highlighted the positive relationship between the declined snowfall and reduced groundwater recharge over Iran (Lines 166-171).

REVIEWERS' COMMENTS

Reviewer #1 (Remarks to the Author):

The authors have given it their best to answer the questions I had. They also performed additional analyses to provide the main causes of groundwater recharge decline, as well as providing additional uncertainty analyses on the slope estimates.

The data are now available through ZENDO, which is important. It is a pity that the coordinates of springs and qanats are not available, but then I urge the authors to also provide the shapefiles of the aquifer boundaries and the names of the aquifers the time series belong to,

In their description of the reasons behind reduced recharge they also mention more efficient irrigation technology leading to more agricultural development because of the additional water becoming available. This is known as the pendulum effect and they could add a reference to that effect if they agree.

Otherwise, I think this paper is ready to be accepted.

Reviewer #2 (Remarks to the Author):

The authors made a great effort to improve the manuscript during the revision.

Reviewer #3 (Remarks to the Author):

Thanks for the revision. The revised paper provides another line of evidence for how unsustainable resource management can cause drastic consequences on the Earth System process. Particularly for the case of Iran, this work can be insightful given the depth and gravity of nation-wide water insecurity. Kudos to authors!

RESPONSE LETTER TO COMMENTS ON “Decline in Iran’s Groundwater Recharge”
MANUSCRIPT NUMBER: NCOMMS-23-07744.R2

Response to Comments of Reviewer #1

General Comment

- The authors have given it their best to answer the questions I had. They also performed additional analyses to provide the main causes of groundwater recharge decline, as well as providing additional uncertainty analyses on the slope estimates. The data are now available through zenodo, which is important. It is a pity that the coordinates of springs and qanats are not available, but then I urge the authors to also provide the shapefiles of the aquifer boundaries and the names of the aquifers the time series belong to.

In their description of the reasons behind reduced recharge, they also mention more efficient irrigation technology leading to more agricultural development because of the additional water becoming available. This is known as the pendulum effect and they could add a reference to that effect if they agree.

Otherwise, I think this paper is ready to be accepted.

- **Response**

We would like to thank Reviewer #1 for their positive evaluation of our work and their valuable feedback.

We have now added the shapefile of the aquifer boundaries and their representative code to the repositated data in <https://zenodo.org/record/8382150>. In addition, we added shapefiles of the basins and primary basins to better support the readers in regeneration of our presented results in the manuscript.

We agree with the comment on the notion of pendulum swing, and we added a relevant reference:

Van Emmerik, T.H.M., Li, Z., Sivapalan, M., Pande, S., Kandasamy, J., Savenije, H.H.G., Chanan, A. and Vigneswaran, S., 2014. Socio-hydrologic modeling to understand and mediate the competition for water between agriculture development and environmental health: Murrumbidgee River basin, Australia. *Hydrology and Earth System Sciences*, 18(10), pp.4239-4259.

Response to Comments of Reviewer #2

General Comment:

- The authors made a great effort to improve the manuscript during the revision.

- **Response**

We would like to thank Reviewer #2 for the positive evaluation of our work and the constructive feedback that helped us to shape our paper well.

Response to Comments of Reviewer #3

General Comment

- Thanks for the revision. The revised paper provides another line of evidence for how unsustainable resource management can cause drastic consequences on the Earth System

process. Particularly for the case of Iran, this work can be insightful given the depth and gravity of nation-wide water insecurity. Kudos to authors!

- **Response**

We would like to thank Reviewer #3 for the positive evaluation of our work and the constructive feedback that helped us to better shape our paper.